# Ligand-specific regulation of transforming growth factor beta superfamily factors by leucine-rich repeats and immunoglobulin-like domains proteins

**Ahmad Abdullah** 📷, **Carl Herdenberg** 📷, **Håkan Hedman** 📷 *

Department of Radiation Sciences, Oncology, Umeå University, Umeå, Sweden

* hakan.hedman@umu.se

**Data Availability Statement:** All relevant data are within the paper and its Supporting Information files.

## Abstract

Leucine-rich repeats and immunoglobulin-like domains (LRIG) are transmembrane proteins shown to promote bone morphogenetic protein (BMP) signaling in *Caenorhabditis elegans*, *Drosophila melanogaster*, and mammals. BMPs comprise a subfamily of the transforming growth factor beta (TGFβ) superfamily, or TGFβ family, of ligands. In mammals, LRIG1 and LRIG3 promote BMP4 signaling. BMP6 signaling, but not BMP9 signaling, is also regulated by LRIG proteins, although the specific contributions of LRIG1, LRIG2, and LRIG3 have not been investigated, nor is it known whether other mammalian TGFβ family members are regulated by LRIG proteins. To address these questions, we took advantage of *Lrig*-null mouse embryonic fibroblasts (MEFs) with doxycycline-inducible *LRIG1*, *LRIG2*, and *LRIG3* alleles, which were stimulated with ligands representing all the major TGFβ family subgroups. By analyzing the signal mediators pSmad1/5 and pSmad3, as well as the induction of *Id1* expression, we showed that LRIG1 promoted BMP2, BMP4, and BMP6 signaling and suppressed GDF7 signaling; LRIG2 promoted BMP2 and BMP4 signaling; and LRIG3 promoted BMP2, BMP4, BMP6, and GDF7 signaling. BMP9 and BMP10 signaling was not regulated by individual LRIG proteins, however, it was enhanced in *Lrig*-null cells. LRIG proteins did not regulate TGFβ1-induced pSmad1/5 signaling, or GDF11- or TGFβ1-induced pSmad3 signaling. Taken together, our results show that some, but not all, TGFβ family ligands are regulated by LRIG proteins and that the three LRIG proteins display differential regulatory effects. LRIG proteins thereby provide regulatory means for the cell to further diversify the signaling outcomes generated by a limited number of TGFβ family ligands and receptors.

## Introduction

The transforming growth factor β (TGFβ) superfamily (hereafter called the TGFβ family) is a family of secreted growth factors that regulate a variety of biological processes, both in development and in adult homeostasis [1,2]. The mature and bioactive TGFβ ligands comprise

**Funding:** This research was supported by the Swedish Cancer Society (https://www.cancerfonden.se), contract number 21 1583 Pj, to HH; the Kempe Foundation (https://www.kempe.com), grant number JCK-1829, to HH; and the Lion's Cancer Research Foundation at Umeå University (https://cancerforskningsfonden.se/), grant numbers LP 21-2276, LS 21-152, and LS 22-153, to HH; and by the regional agreement between Umeå University and Västerbotten County Council on cooperation in the field of Medicine, Odontology and Health, ALF, grant number RV-967035, to HH. The funders had no role in study design, data collection and analysis, decision to publish, or preparation of the manuscript.

**Competing interests:** The authors have declared that no competing interests exist.

homodimers and heterodimers formed by two covalently associated TGFβ family peptides, which in mammals are encoded by 33 different genes [3]. The TGFβ ligand family can be subdivided into four main subfamilies: activins and inhibins, TGFβ factors, bone morphogenetic proteins (BMPs) and growth and differentiation factors (GDFs), and so-called distant members [4]. TGFβ family ligands activate signaling by binding to specific type 1 and type 2 serine/threonine kinase receptors that either couple to the TGFβ-activin signaling branch through phosphorylation of Smad2/3 or to the BMP/GDF signaling branch through phosphorylation of Smad1/5/8 [3–8]. Phosphorylated Smad2/3 and Smad1/5/8 associate with Smad4, forming a complex that accumulates in the nucleus, where it regulates the transcription of target genes [3,9,10].

Leucine-rich repeats and immunoglobulin-like domains (LRIG) constitute a family of transmembrane proteins that in mammals comprises three members, LRIG1 [11,12], LRIG2 [13], and LRIG3 [14]. Recently, we showed that LRIG protein-deficient (*Lrig1-/-;Lrig2-/-;Lrig3-/-*) mouse embryonic fibroblasts (MEFs) display impaired BMP4 and BMP6 signaling [15]. Conversely, TGFβ1 signaling appeared normal in the *Lrig*-deficient cells. In these cells, BMP4 signaling could be rescued by the expression of LRIG1 or LRIG3 but not by the expression of LRIG2. It is not known whether other TGFβ family members are regulated by LRIG proteins or whether the different LRIG proteins have differential effects on the different TGFβ family ligands.

Here, we assessed the role of LRIG proteins in signaling induced by 13 diverse TGFβ family members that represented all the major TGFβ subfamilies and subgroups. We show that the three mammalian LRIG proteins display differential effects on signaling induced by the different TGFβ family ligands.

## Materials and methods

### Growth factors and antibodies

The TGFβ ligands used in this study and their reported median effective dose ($ED_{50}$) are listed in S1 Table. The antibodies used in this study and their dilutions are listed in S2 Table.

### Cell lines and cell culture

Wild-type MEF lines, *Lrig*-null MEF lines, and *Lrig*-null MEFs with doxycycline-inducible *LRIG1*, *LRIG2*, or *LRIG3* alleles have been described previously [15]. The human ovarian carcinoma cell line OVSAHO (cell number: JCRB1046) was obtained from the Japanese Collection of Research Bioresources (Osaka, Japan); the human pre-B acute lymphoblastic leukemia cell line 697 was obtained from the Leibniz Institute DSMZ-German collection of microorganisms and cell cultures (Braunschweig, Germany); the wild-type and *LRIG1*-null SV40 large T-transformed human embryonic kidney cell line HEK293T has been described previously [16]; the hTERT-immortalized human mammary epithelial cell line hTERT-HME1 was obtained from Clontech Laboratories (Palo Alto, CA, USA); and the human ductal breast carcinoma cell line T-47D was obtained from a colleague and authenticated through short tandem repeat profiling via Eurofins Genomics Germany GmbH (Ebersberg, Germany). MEFs, HEK293T[WT], and HEK293T[LRIG1-KO] were cultured in Dulbecco's modified Eagle's medium (Sigma–Aldrich Sweden AB, Stockholm, Sweden) with 10% fetal bovine serum (Fisher Scientific GTF AB, Gothenburg, Sweden), MEM nonessential amino acids (Fisher Scientific GTF AB), 50 μg/ml gentamicin (Invitrogen, Fisher Scientific GTF AB), and 50 μM 2-mercaptoethanol (Sigma–Aldrich Sweden AB). OVSAHO, 697, and T-47D were cultured in RPMI 1640 (Sigma–Aldrich Sweden AB) with 10% fetal bovine serum and 50 μg/ml gentamicin; the 697 culture medium was further supplemented with 110 μg/ml sodium pyruvate (Sigma-Aldrich Sweden AB, #S8636), hTERT-HME1 was cultured in mammary epithelial cell growth basal medium

(MEBM) from BioNordika Sweden AB (Solna, Sweden) supplemented with mammary epithelial cell growth medium from a SingleQuots (MEGM) kit, containing 52 μg/ml bovine pituitary extract, 0.5 μg/ml hydrocortisone, 10 ng/ml epidermal growth factor, 5 μg/ml insulin, 50 μg/ml gentamicin, and 50 ng/ml amphotericin, from Fisher Scientific GTF AB (Lonza #CC-4136). Cells were maintained at 37°C in a humidified atmosphere containing 5% $CO_2$.

## Phospho-Smad1/5 cell immunofluorescence assay

The pSmad1/5 cell immunofluorescence assay was performed essentially as previously described [15] with minor modifications. Briefly, $3x10^3$ cells were seeded per well in a 96-well transparent bottom black well plate (#655090, Greiner Bio-One International GmbH, Monroe, NC, USA) coated with 0.1% bovine gelatin (#G9391, Sigma–Aldrich Sweden AB). For induction of LRIG protein expression, 100 ng/ml doxycycline (Clontech Laboratories, BioNordika Sweden AB, Stockholm, Sweden) was added to LRIG1/2/3-inducible cells after seeding of the cells. The next day, the cells were serum-starved for 1 hour followed by treatment with the ligands for 1 hour. Then, the cells were washed with phosphate-buffered saline (PBS) and fixed with 4% formaldehyde for 10 minutes followed by permeabilization with 0.2% saponin (#S4521, Sigma–Aldrich Sweden AB) for 10 minutes. Next, the cells were blocked with PBS containing 5% fetal bovine serum and 0.1% Tween-20 for 1 hour. Then, the cells were incubated overnight with pSmad1/5 primary antibody at 4°C. The next day, the cells were washed with PBS three times followed by incubation with the secondary antibody (Alexa Fluor 647) for 1 hour. For nuclear staining, cells were incubated with 1 μg/ml Hoechst 33342 stain. The wells were filled with 200 μl of PBS. The plates were imaged, and the fluorescence was quantified in a BioTek Cytation 5 cell image multimode reader (Agilent Technologies, Sweden AB, Sundbyberg, Sweden). The Hoechst-stained nuclei were used as a primary mask for quantifying the pSmad1/5 fluorescence coming from the nuclear regions. The images were processed with the rolling ball subtraction algorithm, and the object mean fluorescence was calculated by the machine's software (Gen5 image prime 3.10). The background fluorescence was subtracted, and graphs were made. Representative fluorescence images used for pSmad1/5 quantification are shown in S1 Fig.

## Immunoblotting

For analysis of the pSmad levels, 30,000 cells were seeded per well in a 24-well plate (#83.3922, Sarstedt AB, Helsingborg, Sweden). For doxycycline-inducible LRIG1/2/3-expressing cells, 100 ng/ml doxycycline was added after seeding the cells. The next day, the cells were serum-starved for 1 hour followed by treatment with ligands for 1 hour. Then, the cells were washed with PBS and lysed with cell lysis buffer (#FNN0011, Invitrogen, Fisher Scientific GTF AB) with protease inhibitor cocktail (#11873580001, Sigma–Aldrich Sweden AB) and PhosSTOP (#4906845001, Roche Diagnostics Scandinavia AB, Bromma, Sweden). Lysates were collected and incubated on ice for 30 minutes and centrifuged at 20,800 x g for 10 minutes at 4°C. The supernatant was collected, and the proteins were separated through sodium-dodecyl-sulfate (SDS) polyacrylamide gel electrophoresis (PAGE) using 10% Bis-Tris gels (#NP0301, Invitrogen, Fisher Scientific GTF AB) followed by transfer onto polyvinylidene fluoride (PVDF) membranes (#1620174, Bio-Rad Laboratories AB, Solna, Sweden). The blots were blocked with 5% bovine serum albumin (#A4503, Sigma–Aldrich Sweden AB) for 1 hour followed by incubation with anti-pSmad1/5, anti-pSmad3, anti-FLAG M2, and anti-actin primary antibodies for 1 hour at room temperature. Then, the blots were washed three times with TBST (20 mM Tris, 150 mM NaCl, 0.1% Tween 20), followed by incubation with fluorescent secondary antibodies for 1 hour at room temperature. Thereafter, the blots were washed three times with

TBST and imaged using an Odyssey CLx imaging system (LI-COR Biosciences GmbH, Bad Homburg, Germany). Image Studio Lite v 5.2 was used for the quantification of the bands.

## RNA isolation and quantitative RT–PCR

For mRNA analysis, cells were seeded in a 12-well plate (#83.3921, Sarstedt AB). For induction of LRIG protein expression in doxycycline-inducible cells, 100 ng/ml doxycycline was added after seeding. The next day, cells were serum-starved for 1 hour followed by treatment with ligands for 1 hour. Cells were washed with PBS and lysed, and total RNA was isolated using a PureLink RNA Mini Kit (#12183018A, Invitrogen, Fisher Scientific GTF AB) containing 1% 2-mercaptoethanol according to the manufacturer's instructions. The expression of specific RNAs was quantified through real-time reverse transcription-polymerase chain reaction (RT–PCR) essentially as previously described [12] but with a CFX96 system C1000 thermal cycler (Bio-Rad Laboratories AB). For *Id1* and *Lrig3* analyses, TaqMan assays Mm00775963_g1 and Mm00622766_m1, respectively, were used (#4331182, Fisher Scientific GTF AB). Primers and probes for *Lrig1*, *Lrig2*, and *Rn18s* are shown in S3 Table. Relative mRNA expression levels were calculated by normalizing the cycle threshold (CT) values of respective mRNA to that of *Rn18s* (ΔCT), and the linear values were calculated and plotted as graphs.

## Statistical analyses

All the data are presented as the average of four independent experiments with standard deviations. Statistical comparisons were performed using a two-way analysis of variance (ANOVA) with Tukey's multiple comparison test or a two-sided Student's *t* test (GraphPad Prism, version 9.0), and a p value of less than 0.05 was considered statistically significant.

## Graphics

Line graphs and column charts were generated in GraphPad Prism, version 9.0, and the corresponding figures were assembled in Microsoft PowerPoint version 2301. The Venn diagram in Fig 6 was created with BioRender.com.

# Results

## TGFβ family ligand responsiveness in wild-type MEFs

For analysis of the responsiveness of MEFs to different TGFβ family ligands, we stimulated wild-type MEFs with 13 different TGFβ family ligands at their reported $ED_{50}$ values (S1 Table) and analyzed the phosphorylation of Smad1/5 and Smad3 through immunoblotting (S2 Fig). Eight of the 13 TGFβ family ligands induced pSmad1/5 or pSmad3 signaling in wild-type MEFs: BMP2, BMP4, BMP6, BMP9, BMP10, and GDF7 activated pSmad1/5 signaling only; GDF11 activated pSmad3 signaling only; and TGFβ1 activated both pSmad1/5 and pSmad3 signaling. BMP3, BMP15, Activin A, GDF3, and GDF15 did not activate pSmad1/5 or pSmad3 signaling at their reported $ED_{50}$ values in wild-type MEFs. Notably, in this initial screen, the ligands were only tested at a single concentration. It is, therefore, possible that some of the apparently inactive ligands would have shown activity at higher concentrations; however, this possibility was not further investigated in the current study.

To investigate the repertoire of expressed TGFβ family receptors and Lrig proteins in wild-type and *Lrig*-null MEFs, we analyzed gene expression data obtained by RNA sequencing in a previous study [15] (S4 Table). The gene expression results suggested that wild-type MEFs and *Lrig*-null MEFs expressed most of the analyzed receptors and coreceptors, except for Acvr1c (also called ALK7), Acvr2b, Amhr2, endoglin, Rgmc (also called Hemojuvelin), and Ror1, for

which the transcript levels were very low or undetectable. There was no significant difference between wild-type and *Lrig*-null cells in regard to their expression levels of TGFβ family type 1 or type 2 receptors or coreceptors. The RNAseq data shown in S4 Table did not discriminate between wild-type and disrupted *Lrig* alleles. To assess the expression of nondisrupted *Lrig* genes in the wild-type and *Lrig*-null MEF populations, we specifically analyzed the expression of the exons that were targeted in the gene ablations via qRT–PCR (S5 Table). This analysis showed that the expression of intact *Lrig* transcripts in the *Lrig*-null MEFs was less than 10% of the levels in the wild-type cells. This result is consistent with a previous demonstration that these *Lrig*-null MEFs do not express detectable levels of Lrig proteins [15].

## LRIG protein regulation of the different TGFβ family ligands

Next, we investigated the capacity of each of the three LRIG proteins to regulate the induction of pSmad1/5 and pSmad3 by stimulatory TGFβ family ligands. To this end, we took advantage of previously established *Lrig*-null MEFs with doxycycline-inducible human *LRIG1*, *LRIG2*, or *LRIG3* alleles. First, to investigate whether the inducible experimental system showed physiological or nonphysiological LRIG protein expression levels, we compared the LRIG1 level in the doxycycline-induced MEFs and the LRIG1 levels in different human cell lines using an anti-human LRIG1 antibody (S3 Fig). We could not directly compare the levels between the inducible *Lrig*-null MEFs and wild-type MEFs because the wild-type MEFs express mouse proteins, whereas the inducible LRIG alleles were of human origin. Immunoblotting showed that LRIG1 could not be detected in the uninduced *Lrig*-null MEFs, whereas doxycycline treatment induced a clear expression of human LRIG1 protein. Importantly, the expression level did not appear to be supraphysiological as it did not significantly differ from the level observed in the human ovarian carcinoma cell line OVSAHO (S3 Fig).

Then, *Lrig*-null MEFs with doxycycline-inducible *LRIG1*, *LRIG2*, or *LRIG3* alleles were treated with stimulatory TGFβ family ligands, and the pSmad1/5 (Fig 1) and pSmad3 (Fig 2) responses were quantified through cell immunofluorescence and immunoblotting, respectively. Although the immunofluorescence pSmad1/5 assay was convenient and allowed high-throughput approaches, immunoblotting was chosen for the pSmad3 analyses because our pSmad3 antibody was not specific enough in the immunofluorescence assay. Hence, BMP2-induced pSmad1/5 signaling was enhanced by LRIG1, LRIG2, and LRIG3. BMP4-induced pSmad1/5 signaling was, as previously reported [15], enhanced by LRIG1 and LRIG3 and enhanced by LRIG2. BMP6-induced pSmad1/5 signaling was enhanced by LRIG1 and LRIG3, whereas the apparent enhancement by LRIG2 was nonsignificant ($p = 0.068$, two-way ANOVA without post-hoc test). GDF7-induced pSmad1/5 signaling showed intriguing LRIG regulation; LRIG1 did not affect sensitivity to low concentrations of GDF7 but reduced the maximal pSmad1/5 response, whereas LRIG3 enhanced pSmad1/5 signaling at low ligand concentrations but did not appear to affect the maximal pSmad1/5 response. BMP9-, BMP10-, and TGFβ1-induced pSmad1/5 signaling (Fig 1) and TGFβ1- and GDF11-induced pSmad3 signaling (Fig 2) were not affected by the expression of LRIG1, LRIG2, or LRIG3.

For analysis of whether the apparently divergent effects of LRIG1 and LRIG3 on GDF7 signaling were quantitative or qualitative in nature, the LRIG1 and LRIG3 expression levels were varied via titrations of the transcription-inducer doxycycline. Then, the correlations between the LRIG protein expression levels and the pSmad1/5 signal level in response to a single concentration of GDF7 (100 ng/ml) were determined (Fig 3). Clearly, LRIG1 and pSmad1/5 showed a negative correlation ($r = -0.8305$), whereas LRIG3 and pSmad1/5 showed a positive correlation ($r = 0.6477$), indicating that LRIG1 suppressed GDF7 signaling, whereas LRIG3 enhanced GDF7 signaling, in a dose-dependent manner.

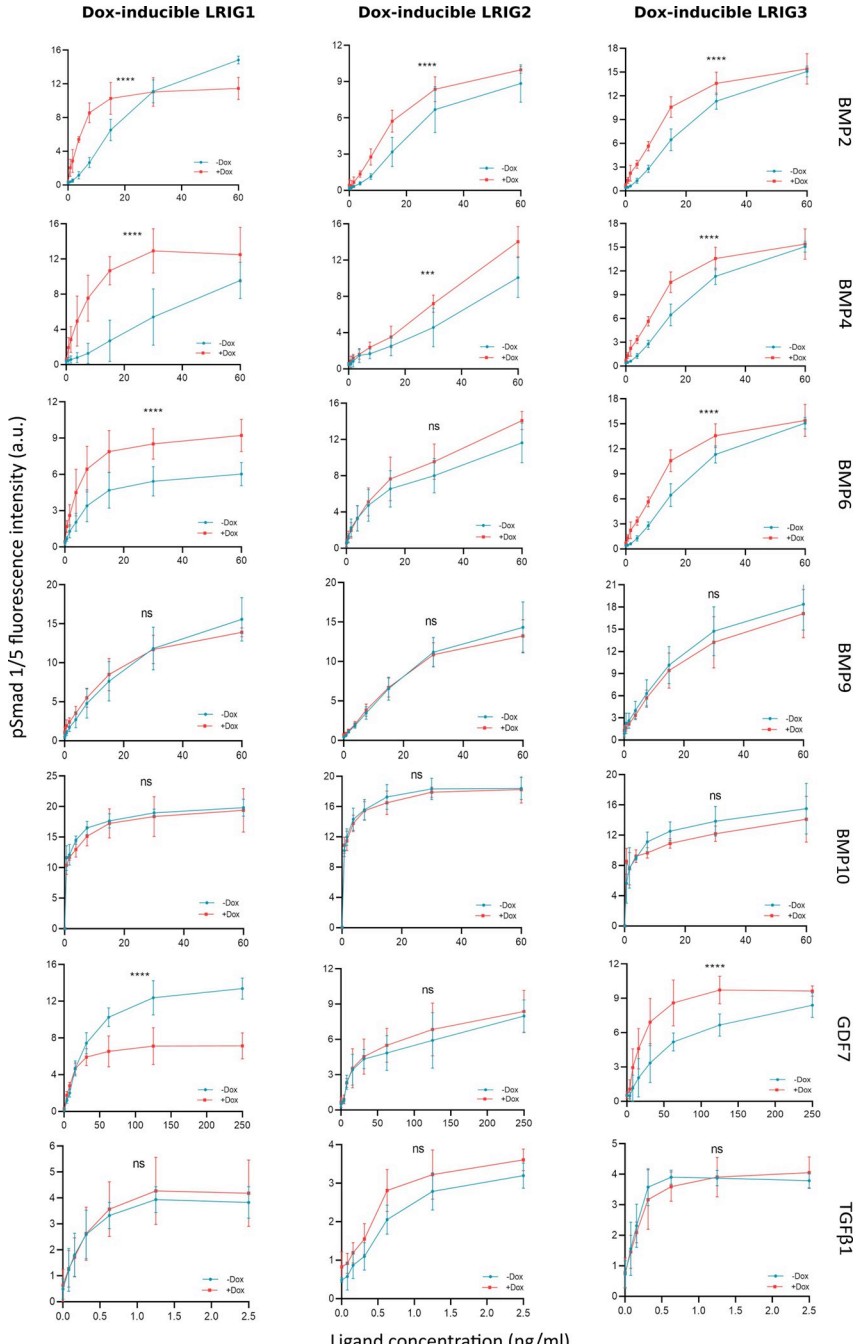

**Fig 1. Phospho-Smad1/5 responses of *Lrig*-null MEFs with doxycycline-inducible *LRIG1*, *LRIG2*, or *LRIG3* alleles to different TGFβ family ligands.** LRIG1, LRIG2, or LRIG3 was induced by treating the cells with 100 ng/ml doxycycline for 24 hours, followed by serum starvation and treatment with various concentrations of TGFβ family ligands for 1 hour. Nuclear pSmad1/5 was quantified via cell immunofluorescence. Graphs show the means of four independent experiments performed in duplicate, with error bars representing the standard deviations between experiments (significant differences between the doxycycline-induced and untreated cells are indicated with asterisks: $^{ns}$p > 0.05; ***p < 0.001; ****p < 0.0001, two-way ANOVA with Tukey's multiple comparison test). Individual graphs are shown in S4 Fig.

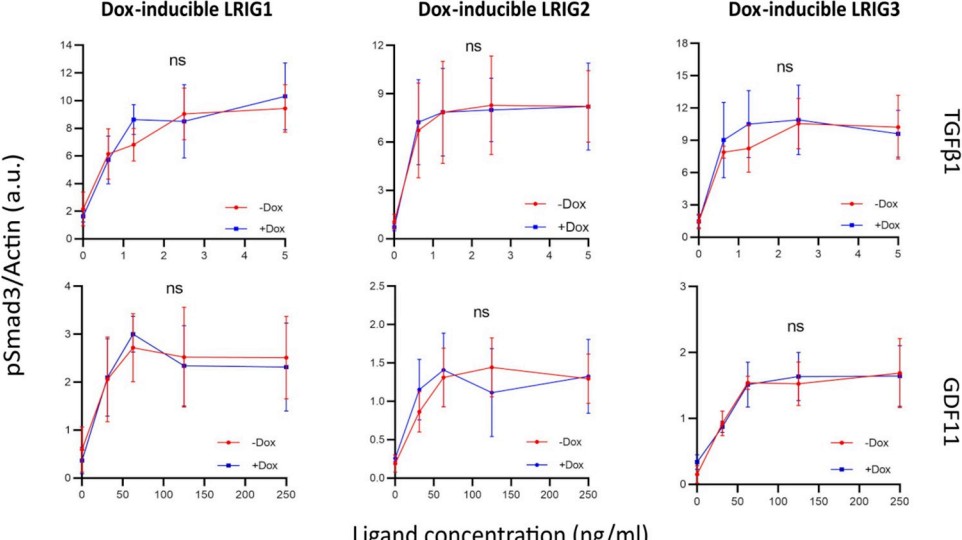

**Fig 2. Phospho-Smad3 responses of *Lrig*-null MEFs with doxycycline-inducible *LRIG1*, *LRIG2*, or *LRIG3* alleles to different TGFβ family ligands.** LRIG1, LRIG2, or LRIG3 was induced by treating the cells with 100 ng/ml doxycycline for 24 hours, followed by serum starvation and treatment with various concentrations of TGFβ family ligands for 1 hour. Relative pSmad3 and actin levels were quantified by immunoblotting. Graphs show the mean pSmad3/actin ratios of four independent experiments, with error bars representing the standard deviations between experiments. There were no significant differences between doxycycline-induced and untreated cells for either of the ligands ($^{ns}p > 0.05$, two-way ANOVA with Tukey's multiple comparison test). Individual graphs are shown in S5 Fig and the full blots of the four experiments are shown in S6 Fig.

## *Lrig*-null MEFs show aberrant BMP and GDF signaling

For analysis of whether endogenous Lrig proteins play a role in the regulation of TGFβ family signaling in MEFs, wild-type and *Lrig*-null MEFs were treated with different concentrations of the seven TGFβ family ligands that induced pSmad1/5 signaling in our pilot screen (S2 Fig). As previously shown, *Lrig*-null MEFs showed impaired BMP4 and BMP6 signaling (Fig 4). Additionally, the *Lrig*-null MEFs showed impaired BMP2 signaling and increased BMP9, BMP10, and GDF7 pSmad1/5 signaling (Fig 4). These results showed that endogenous Lrig proteins were important determinants of TGFβ family signaling in MEFs.

## *Id1* regulation by LRIG proteins

Finally, to investigate whether the observed effects of LRIG proteins on pSmad1/5 signaling translated into the regulation of gene expression, we quantified the expression of the BMP-responsive gene *Id1* by qRT–PCR (Fig 5). In accordance with the pSmad1/5 results, the expression of LRIG1, LRIG2, and LRIG3 enhanced the induction of *Id1* when the cells were stimulated with BMP2 or BMP4, whereas only LRIG1 and LRIG3 enhanced the induction of *Id1* when the cells were treated with BMP6. With respect to GDF7, LRIG1 expression showed an inhibitory effect at the highest GDF7 concentration, whereas LRIG3 enhanced the *Id1* response at both low and high GDF7 concentrations.

## Discussion

Here, we showed that mammalian LRIG proteins regulated TGFβ family signaling differentially and in a ligand-specific manner. Previously, it was shown that the sole LRIG paralogs in the nematode *Caenorhabditis elegans* (SMA-10) and in the insect *Drosophila melanogaster*

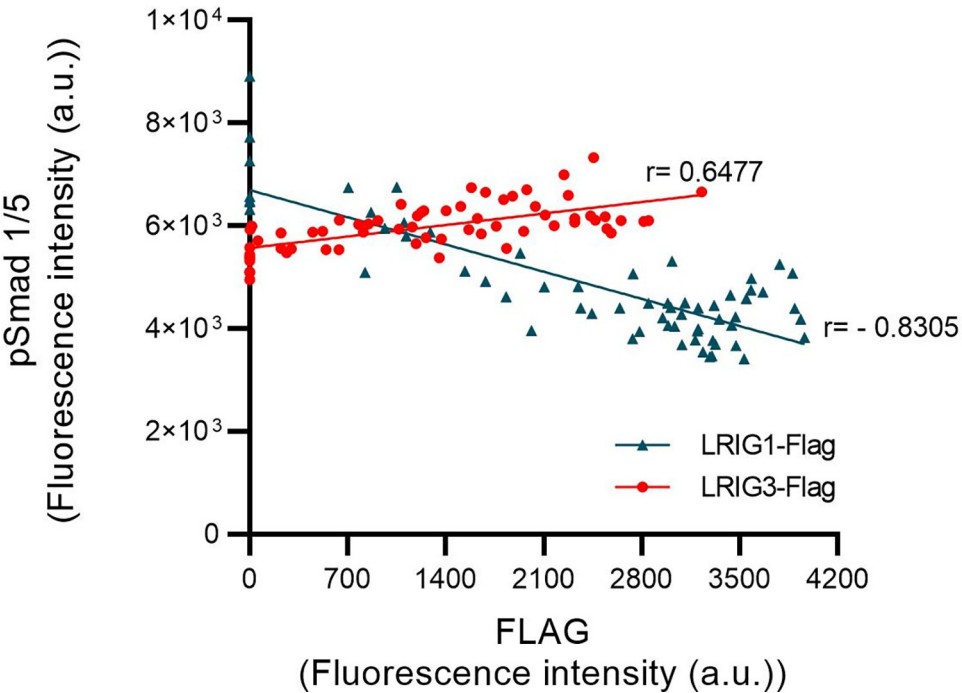

**Fig 3. Phospho-Smad1/5 responses to GDF7 according to LRIG1 and LRIG3 expression levels.** *Lrig*-null MEFs with doxycycline-inducible *LRIG1* or *LRIG3* alleles were induced to express different levels of LRIG proteins by treating the cells with serially diluted concentrations of doxycycline (from 0 to 1000 ng/ml) for 24 hours, followed by serum starvation and treatment with 100 ng/ml GDF7 for 1 hour. Nuclear pSmad1/5 and FLAG expression was quantified via cell immunofluorescence. FLAG immunoreactivity corresponds to relative LRIG1 and LRIG3 expression levels because the doxycycline-inducible *LRIG* alleles encoded FLAG-tagged LRIG proteins. Correlation coefficients (r) are indicated next to the lines in the graph. Every dot in the scatter plot represents an individual value from four independent experiments performed in duplicate. Simple linear regression analysis was performed in GraphPad Prism 9.0, with the fitted lines indicating the relationship. The relationships between LRIG1-FLAG and LRIG3-FLAG immunoreactivities and doxycycline concentrations are shown in S7 Fig.

(Lambik) promote BMP signaling [17,18] and that mammalian LRIG1 and LRIG3 promote BMP4 signaling [15]. Here, we extended these observations to comprise an analysis of the regulatory interactions between the three mammalian LRIG proteins and all the major TGFβ family subgroups. The results showed that LRIG1, LRIG2, and LRIG3 regulated the different branches of mammalian TGFβ signaling in different manners. For example, whereas BMP2 and BMP4 signaling was promoted by all three LRIG proteins, BMP6 signaling appeared to be promoted by LRIG1 and LRIG3 only. Furthermore, GDF7 signaling was oppositely regulated by LRIG1 and LRIG3, since LRIG1 suppressed and LRIG3 promoted GDF7 signaling. Conversely, TGFβ1 and GDF11 signaling was unaffected by LRIG protein expression. Our observations add important information to our understanding of the complex regulation of TGFβ family signaling in mammalian cells.

Is LRIG protein-mediated regulation of TGFβ family signaling physiologically relevant? Indeed, we believe so for several reasons. First, the doxycycline-inducible LRIG expression system used did not produce supraphysiological LRIG expression levels. Rather, the expression of LRIG1 in the LRIG1-inducible MEFs was similar to the expression level in human ovarian carcinoma cells, showing that the expression system produced physiological LRIG1 levels. Second, endogenous MEF Lrig proteins seemed to regulate TGFβ signaling in MEFs because *Lrig* triple knockout (*Lrig*-null) MEFs showed impaired BMP2, BMP4, and BMP6 signaling, altered GDF7 signaling, and enhanced BMP9 and BMP10 signaling. Furthermore, some of the

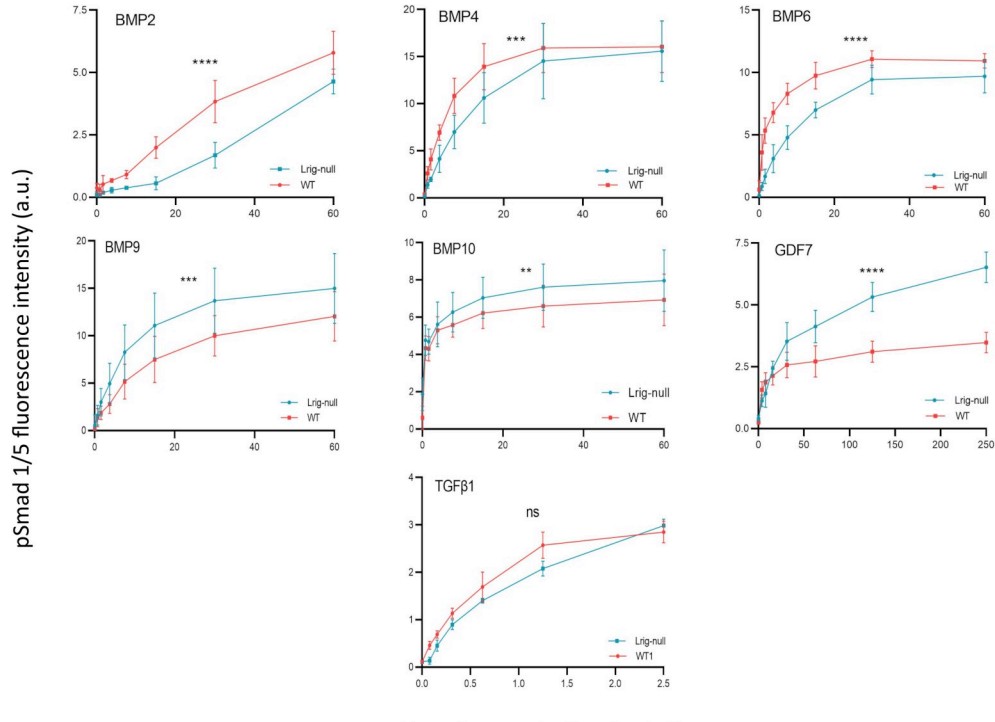

**Fig 4. Phospho-Smad1/5 response of *Lrig*-null MEFs compared with wild-type MEFs to different TGFβ family ligands.** Cells were treated with various concentrations of TGFβ family ligands for 1 hour followed by quantification of nuclear pSmad1/5 via cell immunofluorescence. Graphs show the means of four independent experiments using four different biological replicates of each genotype (four wild-type MEF lines (WT) and four *Lrig*-null MEF lines (*Lrig*-null)) performed in duplicate, with error bars representing the standard deviations between experiments (significant differences between wild-type and *Lrig*-null cells are indicated with asterisks: **p < 0.01; ***p < 0.001; ****p < 0.0001, two-way ANOVA with Tukey's multiple comparison test). Individual graphs are shown in S8 Fig.

phenotypes observed in *Lrig* knockout mice are consistent with effects on TGFβ family signaling. For example, the expansion of certain stem cell compartments observed in *Lrig1*-knockout mice [19] and the craniofacial and inner ear defects seen in *Lrig3*-knockout mice [20] are all consistent with impaired BMP signaling in these *Lrig* knockout mice. The connection between LRIG2 and TGFβ family signaling remains less clear [15,19]; however, here, we showed that LRIG2 can regulate BMP2 and BMP4 signaling, albeit less potently than LRIG1 and LRIG3.

At this stage, the molecular mechanisms behind the signal-regulating functions of LRIG proteins remain unclear. In *Caenorhabditis elegans*, it was suggested that LRIG/SMA-10 promotes BMP signaling via the regulation of BMP type 1 receptor, SMA-6, trafficking [18]. This hypothesis has not been directly addressed in mammalian cells. However, the observed ligand specificities suggest that the LRIG proteins act either on specific receptors or on the ligands themselves. Our results are compatible with both scenarios. In the first scenario, our ligand survey identifies the specific receptors that might be subject to regulation by LRIG proteins (Fig 6). For example, because BMP2/4/6 and BMP9/10 share the same predominant MEF BMP type 2 receptor, Bmpr2, but only BMP2/4/6, and not BMP9/10, are regulated by individual LRIG proteins, Bmpr2 does not seem to be regulated by LRIG proteins. Similarly, because the cells responded in an LRIG-indifferent manner to GDF11, TGFβ1, BMP9, and BMP10, we propose that none of the type 1 receptors, Acvrl1 (also called ALK1), Acvr1 (also called ALK2),

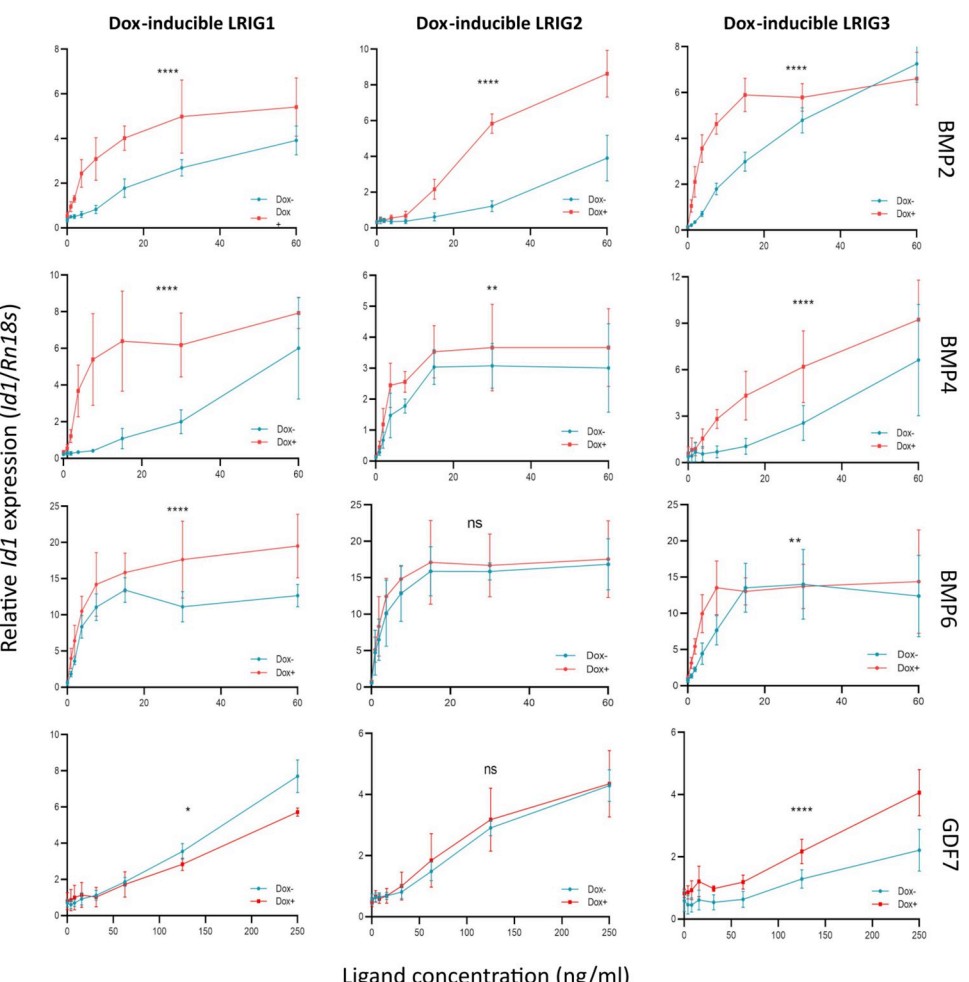

**Fig 5. *Id1* transcriptional responses of *Lrig*-null MEFs with doxycycline-inducible *LRIG1*, *LRIG2*, or *LRIG3* alleles.**
LRIG1, LRIG2, or LRIG3 was induced by treating the respective cell line with 100 ng/ml doxycycline for 24 hours, followed by serum starvation and treatment with various concentrations of TGFβ family ligands for 1 hour. Thereafter, total RNA was isolated, and the relative levels of *Id1* and the reference gene *Rn18s* were quantified through quantitative RT–PCR. Shown are the *Id1*/*Rn18s* ratios in arbitrary units according to TGFβ family ligand concentrations. The graphs represent the average of four individual experiments, with error bars showing the standard deviations between experiments (two-way ANOVA with Tukey's multiple comparison test: $^{ns}p > 0.05$; $^*p < 0.05$; $^{**}p < 0.01$; $^{****}p < 0.0001$). Individual graphs are shown in S9 Fig.

or Tgbr1 (also called ALK5), are targeted by LRIG proteins. However, because GDF7, BMP2, BMP4, and BMP6 are all regulated by LRIG proteins, Bmpr1a (also called ALK3) and Bmpr1b (also called ALK6) are both possible targets for regulation by LRIG proteins. However, it should be noted that the situation might be more complex because although BMP9 and BMP10 signaling was not affected by individual LRIG proteins, their signaling was enhanced in *Lrig*-null cells.

Intriguingly, LRIG1 and LRIG3 regulated GDF7 signaling in opposite directions, i.e., LRIG1 suppressed GDF7 signaling and LRIG3 enhanced GDF7 signaling. Indeed, the ligand specificity shown by LRIG1 is reminiscent of the specificities shown by repulsive guidance molecule (RGM) proteins. Similar to LRIG1, RGM proteins promote signaling induced by BMP2, BMP4, and BMP6 but not signaling induced by BMP9 [25], and RGMB suppresses rather than promotes GDF5 signaling [26]. GDF5 and GDF7 belong to the same subgroup of

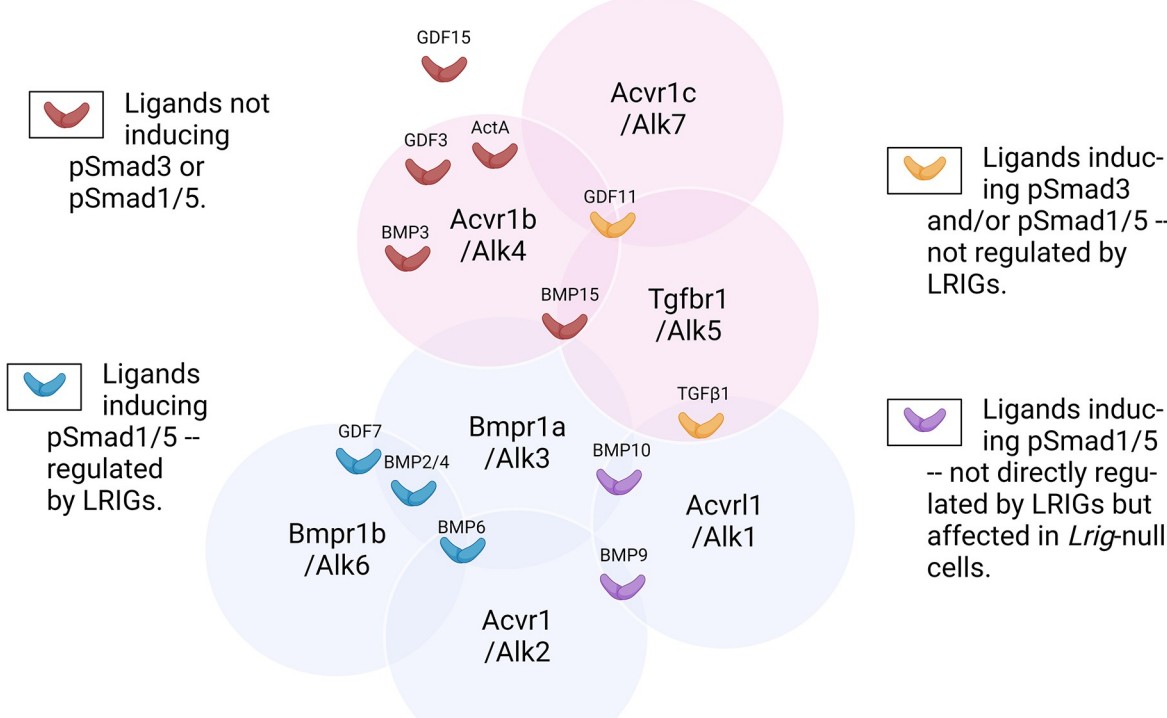

**Fig 6. Venn diagram showing the discovered LRIG protein-mediated regulation of the pSmad responses in MEFs according to the TGFβ family type 1 receptor specificities of the respective ligand.** Ligand–receptor specificities are according to Gipson et al. [7]. GDF15 does not show specific binding to any of the TGFβ family type 1 receptors [21–24]. TGFβ family ligands are color-coded according to the regulation of their phospho-Smad responses by LRIG proteins. Red ligands (GDF3, GDF15, BMP3, BMP15, and ActA) did not induce any pSmad3 or pSmad1/5 responses at their $ED_{50}$ in wild-type MEFs. Blue ligands (GDF7, BMP2, BMP4, and BMP6) were positively or negatively regulated by LRIG proteins regarding their induction of pSmad1/5. Purple ligands (BMP9 and BMP10) were not regulated by individual LRIG proteins, but their responses were enhanced in *Lrig*-null MEFs. Orange ligands (GDF11 and TGFβ1) were not regulated by LRIG proteins regarding their induction of pSmad3 and/or pSmad1/5. Pink circles represent receptors that preferentially signal through Smad2/3 (Acvr1b, Acvr1c, and Tgfbr1), whereas blue circles represent receptors that preferentially signal through Smad1/5/8 (Bmpr1a, Bmpr1b, Acvr1, and Acvrl1). The general outline of the diagram is an adaptation from Gipson et al. [7].

GDFs [7]. Hence, it is possible that LRIG proteins have the same function as RGM proteins or, alternatively, that LRIG proteins are needed for the function of RGM proteins. RGM proteins bind with high affinity to many BMP and GDF ligands [21]; however, it is not known how that binding translates into the promotion of certain signals and the suppression of others [26]. Future experiments will address whether LRIG proteins and RGM proteins regulate TGFβ family signaling via the same, related, or completely different mechanisms.

In summary, our results showed that LRIG1 and LRIG3, and to a lesser extent LRIG2, regulated specific branches of TGFβ family signaling in MEFs. Based on these discoveries, we propose that LRIG proteins provide a previously largely unrecognized means for the cell to regulate and diversify the signaling outcomes generated by TGFβ family ligands.

## Supporting information

**S1 Fig. Representative immunofluorescence images used for the quantification of pSmad1/5 levels in the experiments performed with the Cytation 5.0 instrument.** Alexa Fluor 647-labeled, nuclear localized pSmad1/5 was quantified and used as the readout for the activity

of the BMP/GDF signaling branch.
(PDF)

**S2 Fig. TGFβ family ligand responsiveness in MEFs.** Thirteen TGFβ family ligands, representing all the major TGFβ family subgroups, were evaluated for their ability to induce the phosphorylation of Smad1/5 (pSmad1/5) or Smad3 (pSmad3) in wild-type MEFs at their $ED_{50}$. Ligands and the concentrations used were as follows: BMP2, 10 ng/ml; BMP3, 100 ng/ml; BMP4, 10 ng/ml; BMP6, 6 ng/ml; BMP9, 2 ng/ml; BMP10, 6 ng/ml; BMP15, 10 ng/ml, Activin A, 2 ng/ml; GDF3, 150 ng/ml; GDF7, 100 ng/ml; GDF11, 100 ng/ml; GDF15, 200 ng/ml; and TGFβ1, 0.5 ng/ml. Relative pSmad1/5, pSmad3, and actin levels were quantified through immunoblotting. The bar graphs show the mean pSmad/actin ratios of four independent experiments, with error bars indicating the standard deviations. (Significant differences compared with the untreated control cells are indicated with asterisks: *p < 0.05; **p < 0.01; ***p < 0.001; ****p < 0.0001, Student's *t* test).
(PDF)

**S3 Fig. LRIG1 expression level in LRIG1-inducible MEFs compared to different human cell lines.** LRIG1 expression was induced by treating LRIG1-inducible MEFs with 100 ng/ml doxycycline for 24 hours (Dox+) or LRIG1 was not induced (Dox-). Respective cell cultures were lysed, and 5 μg of protein was analyzed for LRIG1 and actin by immunoblotting. The bar graphs show individual values, means, and standard deviations for LRIG1/actin ratios on an arbitrary scale from four independent experiments. There was no significant difference in the LRIG1/actin ratio between the LRIG1-induced MEFs and the human ovarian carcinoma cell line OVSAHO (p = 0.204, Student's *t* test).
(PDF)

**S4 Fig. Graphs showing the individual plots of the experimental means shown in Fig 1.**
(PDF)

**S5 Fig. Graphs showing the individual plots of the experimental means shown in Fig 2.**
(PDF)

**S6 Fig. Representative full-length blots used for the quantification of the graphs shown in Fig 2.** Phospho-Smad3 responses of *Lrig*-null MEFs with doxycycline-inducible *LRIG1*, *LRIG2*, or *LRIG3* alleles to TGFβ1 and GDF11. LRIG1, LRIG2, or LRIG3 was induced by treating the respective cell line with 100 ng/ml doxycycline for 24 hours, followed by serum starvation and treatment with various concentrations of TGFβ1 or GDF11 for 1 hour.
(PDF)

**S7 Fig. Graphs showing the concentration dependent expression of FLAG tagged LRIG1 and LRIG3 proteins in LRIG-inducible MEFs treated with doxycycline, as shown in Fig 3.**
(PDF)

**S8 Fig. Graphs showing the individual plots of the experimental means shown in Fig 4.**
(PDF)

**S9 Fig. Graphs showing the individual plots of the experimental means shown in Fig 5.**
(PDF)

**S1 Table. TGFβ family ligands used in this study.**
(PDF)

**S2 Table. Antibodies used in this study.**
(PDF)

**S3 Table. Oligonucleotide primer and probe sequences for quantitative real-time RT–PCR.**
(PDF)

**S4 Table. Gene expression data[a] for TGFβ family receptors and coreceptors and Lrig proteins in wild-type and *Lrig*-null MEFs.**
(PDF)

**S5 Table. Gene expression data (qRT–PCR) for Lrig genes in wild-type and *Lrig*-null MEFs.**
(PDF)

**S1 File. Data behind the figures.**
(XLSX)

**S1 Raw images.**
(PDF)

## Acknowledgments

We thank Annika Holmberg for the qRT–PCR analyses and Nitesh Mistry for help with Bio-Tek Cytation 5 cell imaging and analyses.

## Author Contributions

**Conceptualization:** Carl Herdenberg, Håkan Hedman.

**Data curation:** Ahmad Abdullah.

**Formal analysis:** Ahmad Abdullah, Håkan Hedman.

**Funding acquisition:** Håkan Hedman.

**Investigation:** Ahmad Abdullah, Carl Herdenberg.

**Methodology:** Ahmad Abdullah, Carl Herdenberg.

**Project administration:** Håkan Hedman.

**Supervision:** Håkan Hedman.

**Visualization:** Håkan Hedman.

**Writing – original draft:** Ahmad Abdullah, Håkan Hedman.

**Writing – review & editing:** Carl Herdenberg, Håkan Hedman.

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
