## [Decision Letter · Decision Letter 0]

28 Mar 2023

PONE-D-23-05403Ligand-Specific Regulation of Transforming Growth Factor Beta Superfamily Factors by Leucine-rich Repeats and Immunoglobulin-like Domains ProteinsPLOS ONE

Dear Dr. Hedman,

Thank you for submitting your manuscript to PLOS ONE. After careful consideration, we feel that it has merit but does not fully meet PLOS ONE’s publication criteria as it currently stands. Therefore, we invite you to submit a revised version of the manuscript that addresses the points raised during the review process. 

ACADEMIC EDITOR: Both reviewers raised concerns about: 1) the lack of data on expression of the co-receptors following Dox treatment and how this relates to physiological levels, and 2) inappropriate statistical analyses. These two points muse be addressed with additional data and statistical analyses. Changes to the interpretation of results may be required as a result. Please address the more minor concerns as well.

We look forward to receiving your revised manuscript.

Kind regards,

Daniel J. Bernard

Academic Editor

PLOS ONE

Journal Requirements:

In your cover letter, please note whether your blot/gel image data are in Supporting Information or posted at a public data repository, provide the repository URL if relevant, and provide specific details as to which raw blot/gel images, if any, are not available. Email us at plosone@plos.org if you have any questions

Reviewers' comments:

Reviewer's Responses to Questions

**Comments to the Author**

1. Is the manuscript technically sound, and do the data support the conclusions?

Reviewer #1: No

Reviewer #2: Yes

2. Has the statistical analysis been performed appropriately and rigorously? 

Reviewer #1: No

Reviewer #2: No

3. Have the authors made all data underlying the findings in their manuscript fully available?

Reviewer #1: No

Reviewer #2: Yes

4. Is the manuscript presented in an intelligible fashion and written in standard English?

Reviewer #1: Yes

Reviewer #2: Yes

5. Review Comments to the Author

Reviewer #1: In their manuscript, Abdullah et al. the regulation of TGFβ signaling by leucine-rich repeats and immunoglobulin-like domains (LRIG) using Lrig-null mouse embryonic fibroblasts (MEFs) with a doxycycline-inducible LRIG1, LRIG2, or LRIG3 overexpression system. Cells were treated with different TGFβ ligands, after treatment, SMAD1/5/8 and SMAD3 phosphorylation, as well as Id1 expression was assessed.

Major points:

1) The authors do not show LRIG1, 2, and 3 protein expression levels before and after doxycycline-induction in their cell system and its correlation with normal/basal levels of LRIG proteins in wild-type MEFs cells. The data are not robust. SMAD-3 phosphorylation is weak in most of the treatments in wild-type cells, however, there is stimulation in the mutant cells treated with or without doxycycline. The authors should explain this discrepancy to validate the use of these cells to study SMAD phosphorylation dependant on LRIGs proteins.

2) The statistical analysis are not correct in any of the figures. The authors performed t-tests in an experimental design with 2 variables. Two-way ANOVA should be performed.

3) The authors used different concentrations of doxycycline to analyze LRIG1 and LRIG3 roles after GDF7 stimulation, however, the authors do not show these raw data neither how the different doses of doxycycline affect protein levels of LRIG1 or LRIG3 (Fig. 3).

4) It is not clear how robust and reproducible these data are. Experiments should be graphed using individual points, where each point correspond to one independent experiment in all figures.

5)It is not a clear if Id1 stimulation depends on LRIG proteins, there is dose-response effects in absence of doxycycline (Fig. 5).

Minor comments:

1) Colors and font sizes should be consistent across the figures.

2) Protein and gene nomenclature must be modified: proteins should be written in capital letters and genes in lower-case letters and italicized.

3) Graphs and statistical analysis were done with two different version of GraphPad. Please, clarify.

4) Immunofluorescence images are not available, only the quantification is plotted. A supplementary figure needs be added.

Reviewer #2: Abdullah et al. examined signaling of various TGFbeta ligands through the LRIG1-3 co-receptors in murine MEFs. This represents an extension of earlier work by this group. Overall, the manuscript is straightforward and easy to read. Most of my comments are minor in nature. The only ‘major’ concern (#1 below) is the level of expression of the co-receptors when over-expressed, which was not reported (see below). It is possible that additional experiments will be required to address this concern.

Specific comments:

1. The authors do not show expression of the different receptors following Dox treatment. It is important to show the extent to which each receptor was expressed relative to what is seen in wild-type MEFs. The authors imply that the expression level was or might have been supraphysiological (line 238), but they do not show data that speak to this issue. They should. The data in Fig. 3 show (at least for GDF7) that the effect of LRIG1 and LRIG3 on Smad1/5 activation is dependent on the extent of co-receptor expression. We need to see if any of the potentiation (by LRIG3) or attenuation (by LRIG1) occurs when the proteins are expressed at or near physiological levels. The authors claim to address concerns about supraphysiological expression starting on line 238, but this reviewer was not persuaded by their argument. We need to see protein expression and how it relates to what is seen in wild-type MEFs.

2. Line 19: ‘dependent’ here and elsewhere (‘dependency’, line 184, 233) is not the right word. If something is dependent on something else, it requires it. That is not the case here. LRIG impacts, regulates, influences, etc. signaling by some TGFbeta ligands, but in none of the cases reported here did their signaling depend on any of the LRIG proteins. Please replace the word ‘dependent’ or ‘dependency’ with more accurate terminology.

3. Line 38: While there may be 33 genes that encode TGFbeta ligand subunits, there are many heterodimers. Therefore, the number of ligands is likely to be far greater than 33.

4. Line 46: Based on work from the Hill lab, it is more accurate to say that Smad complexes accumulate rather than translocate into the nucleus.

5. Line 119: Here and in the associated figures, why normalize with actin rather than with total (unphosphorylated) Smads?

6. Line 155: Provide references for the reported EC50s. Are these EC50s specific for these ligands in MEFs? It is surprising that 100 ng/ml was used as the EC50 for GDF11, for example. This seems very high. Also, given the receptor expression in these cells, it is surprising that activin A did not stimulate pSmad3. Perhaps 2 ng/ml is too low in these cells. Ideally, the authors should have done dose-responses for all the ligands used here in these cells. In lieu of asking for these data, I think the authors need to do a better job of justifying the concentrations they used (with references). They might also add some text (to the Discussion) acknowledging that they might have been able to assess LRIG regulation of the other ligands had the latter been tested at different (higher?) concentrations.

7. Line 156: There is no description of western blotting for pSmad1/5 in the Methods. At present, western blotting is limited to pSmad3. Please update the Methods.

8. Line 159: The GDF11 data are not convincing here (they are more convincing is subsequent experiments). In Suppl. Fig. 1, only TGFbeta1 convincingly stimulates pSmad3. I don’t see a difference between GDF11 and GDF15 on any of the blots. Also, in the figure, activin should be labeled activin A.

9. Line 170: For the residual Lrig expression, the authors can (and should) look for sequence reads in the floxed exons. If the recombination was complete, there should be no reads for these exons. This information can and should be reported. This seems particularly important for Lrig1.

10. Line 179: What was the rationale for using IF for pSmad1/5 vs. IB for pSmad3? Why different methods for the two?

11. Lines 201-217 (and Figure 5): There is no discussion/mention of the experiment with LRIG2 for BMP2. Also, why wasn’t LRIG2 tested with the other ligands in this experiment?

12. Line 271: I am not convinced that there is a qualitative difference between LRIG2’s effects on BMP2 vs. BMP4. In figure 1, the difference really looks quantitative. I am not certain the correct statistics were used in these experiments. The Methods (line 142) reference t-tests. Given the nature of these experiments (and in Figs. 2, 4, and 5), the authors should be using two-way ANOVAs (or a non-parametric test if the variances are not equal or if they do not log transform their data). In Fig. 1, there is a clear main effect of Dox for BMP4 and BMP6 (maybe even for TGFbeta1). That is, it is clear that LRIG2 is also potentiating their responses as it does for BMP2. The authors should employ the correct statistics and adjust their discussion, as necessary. The correct stats may impact the interpretation of the data in Fig. 4 (BMP9) as well.

13. Line 272-273: The differential effect of LRIG1 and LRIG3 on GDF7 signaling is arguably the most novel and interesting of the study. It is disappointing that no mechanistic insight is provided. According to Fig. 6, GDF7 signals preferentially via ALK6. But ALK6 is expressed at low levels in these cells (indeed, as in most cells). Might LRIG1 and LRIG3 differentially affect GDF7 signaling via ALK3 or even ALK2?

14. Lines 285-286: Is the discussion here based on the relative abundance of these receptors (at least at the mRNA level) in these cells? If so, that should be stated explicitly, as there are no functional data here on the roles of these different type I receptors.

15. Figure 3: Y axis. SMAD is all upper case here but is written as Smad elsewhere. Be consistent.

16. Figure 6: Activin A should be repositioned. It only used ALK4. It does not signal via ALK7.

6. PLOS authors have the option to publish the peer review history of their article (what does this mean?). If published, this will include your full peer review and any attached files.

Reviewer #1: No

Reviewer #2: No

---

## [Author Response · Author response to Decision Letter 0]

2 Jun 2023

For our response to specific reviewer and editor comments, see attached files (Cover Letter/ Response to Reviewers), please.

---

## [Decision Letter · Decision Letter 1]

5 Jul 2023

PONE-D-23-05403R1Ligand-Specific Regulation of Transforming Growth Factor Beta Superfamily Factors by Leucine-rich Repeats and Immunoglobulin-like Domains ProteinsPLOS ONE

Dear Dr. Hedman,

Thank you for submitting your manuscript to PLOS ONE. After careful consideration, we feel that it has merit but does not fully meet PLOS ONE’s publication criteria as it currently stands. Therefore, we invite you to submit a revised version of the manuscript that addresses the points raised during the review process.

ACADEMIC EDITOR:

The authors were responsive to previous feedback. There are just a few minor issues remaining to be addressed. Please see the reviewer comments. 

We look forward to receiving your revised manuscript.

Kind regards,

Daniel J. Bernard

Academic Editor

PLOS ONE

Journal Requirements:

Reviewers' comments:

Reviewer's Responses to Questions

**Comments to the Author**

1. If the authors have adequately addressed your comments raised in a previous round of review and you feel that this manuscript is now acceptable for publication, you may indicate that here to bypass the “Comments to the Author” section, enter your conflict of interest statement in the “Confidential to Editor” section, and submit your "Accept" recommendation.

Reviewer #1: (No Response)

Reviewer #2: (No Response)

2. Is the manuscript technically sound, and do the data support the conclusions?

Reviewer #1: Partly

Reviewer #2: Yes

3. Has the statistical analysis been performed appropriately and rigorously? 

Reviewer #1: No

Reviewer #2: Yes

4. Have the authors made all data underlying the findings in their manuscript fully available?

Reviewer #1: Yes

Reviewer #2: Yes

5. Is the manuscript presented in an intelligible fashion and written in standard English?

Reviewer #1: Yes

Reviewer #2: Yes

6. Review Comments to the Author

Reviewer #1: In their revised manuscript, Abdullah et al. addressed most of the previous comments from this reviewer. However, the authors are missing some relevant information in the ms.

1) Post-hoc comparisons used in their statistical analysis (two-way ANOVA) needs to be added.

2) Not all the supplementary figures (S1-S9) are discussed and referred in the text. For example, references to each supplementary figure must be added in lines 211, 218, and 286.

Reviewer #2: The authors were highly responsive to my prior feedback. It is unfortunate that the results still lack mechanistic insight, but the basic phenomena are interesting and worth reporting.

I have a few remaining minor comments:

1. There are 9 supplementary figures. These are not presented in order (i.e., reference is made to supplementary figure 9 before 1; 3 is referenced before 2) or are not referenced at all (supplementary figures 4, 5, 6, 7, and 8). Ideally, all supplementary figures should be referenced in the text and/or in the legends, and should be referenced in order of presentation.

2. Line 82: ‘college’ should be ‘colleague’

3. Lines 214-218: no figure is referenced in association with the described data. I think this may relate to supplementary figure 7 (though see my first comment).

7. PLOS authors have the option to publish the peer review history of their article (what does this mean?). If published, this will include your full peer review and any attached files.

Reviewer #1: No

Reviewer #2: No

---

## [Author Response · Author response to Decision Letter 1]

7 Jul 2023

Below follows a point-by-point rebuttal to all the Academic Editor’s and reviewers’ comments, with the comments displayed in italics and our responses (A1-A8) in standard font. 

ACADEMIC EDITOR:

The authors were responsive to previous feedback. There are just a few minor issues remaining to be addressed. Please see the reviewer comments. 

A1: We are pleased to note that only a few minor issues remain to be addressed. 

Reviewer #1: In their revised manuscript, Abdullah et al. addressed most of the previous comments from this reviewer. However, the authors are missing some relevant information in the ms.

A2: We are glad that Reviewer #1 was mostly happy with our revised manuscript. 

1) Post-hoc comparisons used in their statistical analysis (two-way ANOVA) needs to be added.

A3: We thank Reviewer #1 for their statistical advice. Accordingly, we performed Tukey’s multiple comparison test post-hoc to all our two-way ANOVAs. These post-hoc tests did not change any of the significance levels indicated in the previous manuscript. Information about the post-hoc tests were added in the revised manuscript (Lines 165, 230, 458, 469, 495, and 506). 

2) Not all the supplementary figures (S1-S9) are discussed and referred in the text. For example, references to each supplementary figure must be added in lines 211, 218, and 286.

A4: We apologize for this mistake. In the revised manuscript, we referred to all supplementary figures: S1, Line 122; S2, Lines 180 and 250; S3, Lines 212 and 219; S4, Line 459; S5, Line 470; S6, Line 471; S7, Line 485; S8, Line 496; S9, Line 507. We also changed their numbering and order to reflect their order of appearance in the text. 

Reviewer #2: The authors were highly responsive to my prior feedback. It is unfortunate that the results still lack mechanistic insight, but the basic phenomena are interesting and worth reporting.

A5: We are pleased to note Reviewer #2’s positive view on the manuscript. 

I have a few remaining minor comments:

1. There are 9 supplementary figures. These are not presented in order (i.e., reference is made to supplementary figure 9 before 1; 3 is referenced before 2) or are not referenced at all (supplementary figures 4, 5, 6, 7, and 8). Ideally, all supplementary figures should be referenced in the text and/or in the legends, and should be referenced in order of presentation.

A6: We apologize for this mistake. In the revised manuscript we referred to all supplementary figures and changed their numbering and order to reflect their order of appearance in the text. 

2. Line 82: ‘college’ should be ‘colleague’

A7: We thank Reviewer #2 for their careful reading of the manuscript. Accordingly, ‘college’ was changed to ‘colleague’ (Line 82). 

3. Lines 214-218: no figure is referenced in association with the described data. I think this may relate to supplementary figure 7 (though see my first comment).

A8: We apologize for this mistake. In the revised manuscript, we added a reference to Supplementary Figure 3 (Lines 212 and 219).

---

## [Editor Report · Decision Letter 2]

26 Jul 2023

Ligand-Specific Regulation of Transforming Growth Factor Beta Superfamily Factors by Leucine-rich Repeats and Immunoglobulin-like Domains Proteins

PONE-D-23-05403R2

Dear Dr. Hedman,

We’re pleased to inform you that your manuscript has been judged scientifically suitable for publication and will be formally accepted for publication once it meets all outstanding technical requirements.

Kind regards,

Daniel J. Bernard

Academic Editor

PLOS ONE

---

## [Editor Report · Acceptance letter]

11 Aug 2023

PONE-D-23-05403R2 

Ligand-Specific Regulation of Transforming Growth Factor Beta Superfamily Factors by Leucine-Rich Repeats and Immunoglobulin-Like Domains Proteins 

Dear Dr. Hedman:

I'm pleased to inform you that your manuscript has been deemed suitable for publication in PLOS ONE. Congratulations! Your manuscript is now with our production department. 

Kind regards, 

on behalf of

Dr. Daniel J. Bernard 

Academic Editor

PLOS ONE